# Migration of Myogenic Cells Is Highly Influenced by Cytoskeletal Septin7

**DOI:** 10.3390/cells12141825

**Published:** 2023-07-11

**Authors:** Zsolt Ráduly, László Szabó, Beatrix Dienes, Péter Szentesi, Ágnes Viktória Bana, Tibor Hajdú, Endre Kókai, Csaba Hegedűs, László Csernoch, Mónika Gönczi

**Affiliations:** 1Department of Physiology, Faculty of Medicine, University of Debrecen, 4032 Debrecen, Hungary; raduly.zsolt@med.unideb.hu (Z.R.); laszlo.szabo@med.unideb.hu (L.S.); dienes.beatrix@med.unideb.hu (B.D.); szentesi.peter@med.unideb.hu (P.S.); banaagnes21@gmail.com (Á.V.B.); csl@edu.unideb.hu (L.C.); 2ELKH-DE Cell Physiology Research Group, University of Debrecen, 4032 Debrecen, Hungary; 3Doctoral School of Molecular Medicine, University of Debrecen, 4032 Debrecen, Hungary; 4Department of Anatomy, Histology and Embryology, Faculty of Medicine, University of Debrecen, 4032 Debrecen, Hungary; hajdu.tibor@med.unideb.hu; 5Department of Medical Chemistry, Faculty of Medicine, University of Debrecen, 4032 Debrecen, Hungary; ekokai@med.unideb.hu (E.K.); hcsaba@med.unideb.hu (C.H.)

**Keywords:** Septin7, cytoskeleton, migration, myogenesis, regeneration, intracellular calcium, forchlorfenuron

## Abstract

Septin7 as a unique member of the GTP binding protein family, is widely expressed in the eukaryotic cells and considered to be essential in the formation of hetero-oligomeric septin complexes. As a cytoskeletal component, Septin7 is involved in many important cellular processes. However, its contribution in striated muscle physiology is poorly described. In skeletal muscle, a highly orchestrated process of migration is crucial in the development of functional fibers and in regeneration. Here, we describe the pronounced appearance of Septin7 filaments and a continuous change of Septin7 protein architecture during the migration of myogenic cells. In Septin7 knockdown C2C12 cultures, the basic parameters of migration are significantly different, and the intracellular calcium concentration change in migrating cells are lower compared to that of scrambled cultures. Using a plant cytokinin, forchlorfenuron, to dampen septin dynamics, the altered behavior of the migrating cells is described, where Septin7-depleted cells are more resistant to the treatment. These results indicate the functional relevance of Septin7 in the migration of myoblasts, implying its contribution to muscle myogenesis and regeneration.

## 1. Introduction

Cell migration is a critical process within a multicellular organism to regulate the optional development and function of an organ. The mechanism of migration has been studied in several physiological and pathological processes, including embryonic development, wound healing, regeneration, immune response, tumor progression, or metastasis formation [1]. The ability to regenerate has particular importance in skeletal muscle, the tissue that represents 30–40% of total body mass [2], and has an immense capacity for regeneration following injury. Regardless, the causative of changes in skeletal muscle mass and size (age, different diseases, immobilization, a special environment such as microgravity, or simple exercise), muscle stem cells (MuSCs) or satellite cells are activated and they actively contribute to the formation of new fibers through a very complex molecular program [3]. Activated MuSCs migrate from their original place called a niche into the area of injury and with high proliferation capacity create myoblasts, and later multinucleated myotubes and muscle fibers [4]. However, satellite cells are not the only players which contribute to skeletal muscle repair; different circulating progenitors, such as bone-marrow-derived cells could promote myogenesis or secrete paracrine factors affecting the surrounding tissues [5]. During the regeneration process, blood vessels and nerves also infiltrate the wounded area [5]. As the injured area is often localized far away from the stock of the previously mentioned cell populations, their migration capacity is crucial for tissue repair.

In skeletal muscle, precisely organized signaling events regulate the re-organization of the key components of the cytoskeleton and the formation of cell adhesions, processes that drive the myoblasts toward the area of injury [4,6]. The cytoskeleton is a complex system containing several building elements, including microtubules, intermediate filaments, microfilaments and the fourth component of the cytoskeleton called septins [7]. Microtubules determine cell shape and have an important role in directing the movement of cells [8], while intermediate filaments are responsible for mechanical stability of the cells [7]. In myoblast migration, the main identified player so far is actin, which organizes into more complex actin-rich structures (filopodia, lamellipodia, stress fibers, or podosomes) [9]. Within the stress fibers, different factors such as the Rho-family small GTPases (Rho1, Cdc42, and Rac1) and Arp2/3 have an essential role in the regulation of subcellular actin assembly or in the formation of filament branching [9], respectively. Moreover, several studies suggest an interaction between the aforementioned proteins and cytoskeletal septins [10,11].

Septins are guanosine triphosphate (GTP) binding proteins and belong to a highly conserved protein family in eukaryotes [7]. Septins have a role in several physiological and pathological cellular processes, including carcinogenesis, exocytosis, endocytosis, cell division [12,13]. So far, 13 septin genes have been identified in humans; the encoded proteins were classified into four groups (SEPT2, SEPT3, SEPT6, and SEPT7) based on sequence homology [7]. Septin filament assembly and disassembly involves nucleotide binding and hydrolysis; however, the exact underlying mechanism is still poorly understood [14,15]. The role of septins in migration has been investigated in epithelia, fibroblasts, lymphocytes, and neurons [16]. Septin-deficient T lymphocytes are characterized by disorganized migration due to uncontrolled cytoskeletal forces and the severely modified compression and rigidity of membrane, which is essential for the efficient motion and transmigration capacity of these cells [17]. Abnormalities in septin expression could enhance the migratory and invasive properties of cancer cells [16,18,19,20]. Alternative splicing variants of the individual septins could have contrary effects on cell migration, as has been proven with Septin9 [21]. In breast cancer cell lines, Sept9_i1 supported cell migration, while with Sept9_i2 this effect was not detected [21]. Another study on Septin9 showed that even tissue-specific interacting partners could regulate and coordinate the formation of the septin–actin networks [10]. Cell division control protein Cdc42EP5/Borg3 associates with actin structures and regulates actomyosin contractility via Septin9 protein. This biological process helps the amoeboid melanoma cells to migrate [10]. Moreover, the overexpression of Septin6 facilitated hepatocellular carcinoma cell proliferation, migration and invasion and its expression is relatively higher in these tissues [22]. In human non-small-cell lung cancer lines, researchers verified that regulating proteins, e.g., lysine-specific histone demethylase 1, regulate tumor metastasis via demethylating the *SEPT6* promoter, and the result is overexpressed Septin6, which activates the TGF-β1 pathway [23].

On the other hand, the behavior of different septins could be considered controversial. As the expression of Septin7 in various glioma cell lines was lower than that in normal brain cells, a research group revealed that the overexpression of Septin7 significantly inhibited LN18 cell migration and chemotaxis, which was induced by IGF-1 [24]. It was also proven that increased Septin7 expression could depolymerize actin filaments, thus obstructing glioma cell migration [24]. Moreover, even microRNA regulation pathways verified that *SEPT7* could be a tumor suppressor gene in glioblastoma cells [25]. On the contrary, reducing the expression of Septin2 and Septin7 resulted in reduced proliferative migration, and invasion in human breast cancer cell lines [20]. Furthermore, in human osteosarcoma U2-OS cells, Septin7 can maintain cell migration by modulating microtubule nucleation; thus, reduced Septin7 expression decreased the migration of these cells [26]. A research group revealed that the knockdown of Septin7 in primary mouse cardiac endothelial cells reduced persistent directional migration [11].

Forchlorfenuron (FCF; N-(2-Chloro-4-pyridyl)-N9- phenylurea) is a synthetic plant cytokinin that consists of a chlorinated pyridine and a phenol ring joined together by a urea group. The effect of FCF was first investigated in the budding yeast *Saccharomyces cerevisiae*, where the small molecule compound reversibly affected septin filament assembly, and also the localization and morphology of septin complexes [27]. Similar findings were published in a model fungus *Ashbya gossypii* [28], where FCF induced the formation of long, cortically associated septin fibers along with a severe change in hypha stability and cell morphology. While the specificity of FCF is questionable according to recent findings [14,29], it seems that the drug does not modify in vitro either actin or tubulin polymerization in MDCK and HeLa S3 cell lines [30]. So far, it is the only agent which is now available for investigating septin dynamics. In addition, based on its high degree of temporal control over septin function, FCF analogues are generated via the structural modification of the original compound, and these molecules are under investigation in gynecological anti-cancer therapies [31]. It is also well-documented, that changes in the intracellular calcium level are an essential regulatory factor in determining the redistribution of the cytoskeleton and for this reason change the motility and migration of different cell types. An elevated intracellular calcium level can affect mRNA stabilization and cell migration in non-excitable cells, as in primary human monocytes [32], or in epithelial keratinocytes [33]. Specific high-calcium microdomains were described in migrating embryonic lung fibroblasts regulating the complex cellular processes [34]. In excitable cell types such as myoblast culture, different patterns of changes in [Ca^2+^]_i_ were observed during in vitro myogenesis [35]. In human neural progenitor cells, Orai channels are required for calcium homeostasis and they are regulated by Septin7 [36,37]. In addition, several voltage- and stretch-activated ion channels (L-type calcium channel, IP_3_R, TRPC1, and TRPM7) are also identified as key players determining intracellular calcium changes in migrating myoblasts or neurons [34,38,39,40].

According to recent findings, septins and especially Septin7 might have a pivotal role in skeletal muscle physiology and pathology [41]. Here, we present the role of Septin7 in migration using the C2C12 cell line. We demonstrate the basic differences between migrating and non-migrating myoblasts, using a 2D migration assay and live cell imaging experiment. Furthermore, we provide evidence that Septin7 might have a role in regulating intracellular calcium ion concentrations during migration.

## 2. Materials and Methods

### 2.1. Cell Culture

The mouse immortalized C2C12 myoblast cell line was cultured in high-glucose Dulbecco’s modified eagle medium (DMEM) (Biosera, Nuaille, France) supplemented with 10% fetal bovine serum (FBS) (Gibco by Life Technologies, Carlsbad, CA, USA), 1% Penicillin–Streptomycin (Gibco), and 1% L-glutamine (Biosera), at 37 °C, in a 5% CO_2_ thermostated environment. The medium was changed every other day and cells were subcultured at 80–90% confluence. Myoblast fusion and the generation of differentiated myotubes were induced by exchanging the culture media to DMEM containing 2% horse serum (Gibco) at approximately 90% confluence.

### 2.2. Gene Silencing

The modification of Septin7 expression in C2C12 cells was evaluated using the plasmid-based shRNA technique as described earlier [41]. Briefly, C2C12 myoblasts were transfected with Septin7-specific shRNA constructs in retroviral pGFP-V-RS vectors (Origene, Cambridge, UK) using Lipofectamine 2000 transfection reagent in serum-free OptiMEM (Life Technologies). For controls, an ineffective scrambled shRNA cassette vector was employed (Scr). Briefly, 48 h after transfection, a Puromycin (2 µg/mL)-containing medium was applied to select cells containing the specific shRNA sequences. Individual clones were then separated and cultured further to analyze the effective gene silencing via Western blotting. Appropriate clones presenting a significant Septin7 expression change were tested throughout increasing passage numbers, and continuously detectable lower Septin7 expression was required to use the cell clone for further investigation (S7-KD cells).

### 2.3. Immunofluorescent Staining of Cultured Cells

C2C12 cells were subjected to immunocytochemistry as previously described [41]. Briefly, cells were seeded on 30 mm glass coverslips and were fixed with 4% PFA for 15 min. Additionally, 0.1 M glycine in PBS was used to neutralize excess formaldehyde after fixation. Cells were permeabilized with 0.25% Triton-X (TritonX-100, Sigma, St. Louis, MO, USA) for 10 min and blocked with a serum-free protein-blocking solution (DAKO, Los Altos, CA, USA) for 30 min at room temperature. A primary Septin7 antibody (JP18991, IBL, Männedorf, Switzerland) was diluted in the blocking solution and added to the samples, and the slides were incubated overnight at 4 °C in a humidity chamber. On the next day, the samples were washed three times with PBS and incubated with fluorophore-conjugated secondary antibodies and FITC-phalloidin (1:1000) or TRITC-phalloidin (1:1000) for 1 h at room temperature in the dark. After washing three times with PBS, a drop of the mounting medium was added to each slide. Images from Alexa Fluo 488-, TRITC-, FITC-, Hoechst33342- and DAPI-labeled samples were acquired with an AiryScan 880 laser scanning confocal microscope (Zeiss, Oberkocken, Germany) equipped with 20× air, 40× oil and 63× oil objectives. Excitation at 488 nm for Alexa Fluo488, 543 nm for mCherry, and 405 nm for DAPI were used to detect the fluorescence of the secondary antibodies and the fluorescence of the Septin7-N-mCherry protein, while emission was recorded with properly selected wavelength filters for Alexa Fluo488 with BP 495-550, for mCherry with BP 570-620 and for DAPI with BP 420-445. The quantification of changes in filament structure was evaluated by determining the thickness of Septin7 filaments in the fluorescent images of control C2C12 cultures. Zen 3.5 Blue software was used for image processing where five regions within individual cells were randomly selected.

### 2.4. Determination of Cellular Proliferation

The proliferation rate of cultured muscle C2C12 cells was determined using CyQUANT NF Cell Proliferation Assay Kit (Invitrogen, Waltham, MA, USA). C2C12 cells (2500 cells/well) were cultured in 96-well black plates with clear bottoms (Greiner Bio-One, Mosonmagyaróvár, Hungary) for 24 h, and then cells were treated with different concentrations of Forchlorfenuron (Sigma) for 24 h. HBSS buffer was prepared (Component C) with deionized water, and then a once-diluted (1×) dye-binding solution was added to the CyQUANT NF dye reagent (Component A). The growth medium was exchanged to 100 μL of the 1× dye binding solution. The microplate was covered and incubated at 37 °C for 30 min, and fluorescence was measured at 485 nm excitation and 530 nm emission wavelengths using the FlexStation 3 multimode microplate reader (Molecular Devices, San Jose, CA, USA). Relative fluorescence values were expressed as the percentage of non-treated cells regarded as 100%.

### 2.5. Fusion Protein Design and Expression

The Septin7-N-mCherry and Septin7-N-ECFP sequences were cloned into a pcDNA 3.1 (+) expression vector. Cells were transfected with the coding plasmids (either mCherry or ECFP) using the Lipofectamine 2000 transfection reagent at 50–70% confluence in serum-free OptiMEM. Three hours after transfection, the medium was replaced by complete DMEM, and cells were allowed to recover and synthesize the coded protein for 24 or 48 h. A Hoechst 33342 (Sigma) dye solution was used for nucleus staining in living cells. Images were acquired with an AiryScan 880 laser scanning confocal microscope as described above.

### 2.6. Examination of the Septin7-N-mCherry Fusion Protein Expression in Living Cells

C2C12 cells were seeded in 96-well Cell Carrier Ultra plates (6055302, Perkin Elmer, Waltham, MA, USA) in DMEM containing 10% FBS. At 70% confluence, the medium was replaced by serum-free OptiMEM and the cells were transfected with the Septin7-N-mCherry coding plasmid. The Cells were allowed to recover and synthesize the coded protein for 24 h. Then, a cell-free zone was created by a Tecan Freedom EVO liquid handling robot, using the liquid handling arm, with a 10 µL pipette tip. Images were acquired on Opera Phenix High Content Confocal System (Perkin Elmer, Waltham, MA, USA). A total of 24 fields with 200–250 cells were acquired per well and laser-based autofocus was performed for each imaging position. Images of brightfield and Alexa-647 channels were collected at the 5 μm position of the Z image plane relative to the bottom of the optical plate using a 63× water immersion objective (NA: 1.15). 

In order to visualize the cells and the localization of Septin7-N-mCherry protein excitation wavelength 561 nm was used to detect fluorescence, while emission was collected with a properly selected wavelength filter for mCherry with BP 570-630. The cells were recorded every five minutes for 6 h in the time series mode at 37 °C, in a 5% CO_2_ thermostated environment.

### 2.7. Migration Assay and Analysis

Cells (2 × 10^4^) were plated on a 4-well plate (Nunc multidishes; Thermo Scientific) within a silicon insert (Ibidi GmbH, Gräfelfing, Germany) and allowed to adhere to the surface for 24 h. Before the removal of the insert, Mitomycin C (Sigma) treatment was applied (10 ug/uL, 2 h) to block cell proliferation. PBS was applied to wash out Mitomycin C, and then Forchlorfenuron (FCF, 100 μM) or the solvent control (ethanol, Sigma) was used in the media during the experiments. Migration followed at 37 °C in a 5% CO_2_ thermostated environment using CytoSMART^TM^ System. Pictures were taken every 5 min and the cell-free zone was monitored. The magnification of the system was 20×. The images recorded by using CytoSMART^TM^ System (CytoSMART Technologies, Eindhoven, The Netherlands; Lonza Bioscience, Basel, Switzerland) were individually saved from the videos and were further analyzed using ImageJ analyzing software (latest version 1.53t, National Institutes of Health, Bethesda, MD, USA, https://imagej.nih.gov/ij/; accessed on 1 July 2022).

The MathLab-based Cell Tracker Image processing software [42] was used to analyze the migration of individual cells. The length of the total way of run, maximum distance from origin, and average cell speed parameters were calculated, and the migration tracks of the individual cells were also visualized. The number of occasions when cells did not move was calculated during the 12 h of measurement. The movement of an individual cell in any direction was determined within 20 min sections from the original recordings. The cells were considered to be moving if their displacement exceeded their diameter in any direction; otherwise, they were considered to be non-moving.

### 2.8. Measurement of Intracellular Ca^2+^ Concentration ([Ca^2+^]_i_) in Migrating Cells

A wound scratch assay was used to determine the total [Ca^2+^]_i_ of the cells during migration. Briefly, 8 × 10^4^ cells were plated on 30 mm glass coverslips and kept in proliferation media until confluence was reached. Cell-free zones were created by scratching the cell layer with a P200 pipette tip. After 5 h of incubation at 37 °C, in 5% CO_2_, and in serum-free proliferation media, cells were loaded with 2.5 µM Fura-2 AM (Invitrogen, MA, USA) for 20 min at 37 °C. 

After loading the cells were kept in Tyrode solution (in mM: 137 NaCl, 5.4 KCl, 0.5 MgCl_2_, 1.8 CaCl_2_, 11.8 Hepes, and 1 g L^−1^ glucose; pH 7.4). Fura-2 was excited with a CoolLED pE-340fura light source (CoolLED LTD, Andover, England) mounted on a ZEISS Axiovert 200 m inverted microscope (Zeiss, Oberkochen, Germany). The excitation wavelength alternated between 340 and 380 nm, the emission was detected with a 505–570 nm band-pass filter, and measurements were carried out at room temperature. The image acquisition and post-processing were carried out with the AxioVision (rel. 4.8) software (Zeiss, Oberkochen, Germany). The fluorescent ratio representing [Ca^2+^]_i_ was calculated from the images taken at 340 and 380 nm after background correction. In these experiments, two objectives were used, 10× (air) for data collection and 40× (oil) for representative images.

### 2.9. Quantification and Statistical Analysis

Pooled data were expressed as mean ± standard error of the mean (SEM). The differences between data groups were assessed using an ordinary one-way ANOVA, Bonferroni’s post hoc multiple comparison test (GraphPad Software (latest version 9.5.1), San Diego, CA, USA) and Student’s t-test, with a *p* value of less than 0.05 being considered statistically significant.

## 3. Results

### 3.1. Intracellular Appearance and Structure of Septin7 Are Different in Migrating and Non-Migrating Myoblasts

Septin7 is a cytoskeletal protein and plays an essential role in several physiological processes. In our previous report [41], we showed that Septin7 and actin filaments co-localize, and a significant alteration of cell morphology in Septin7-downregulated (knockdown, KD) C2C12 cultures was also detailed. However, the structural reorganization of the Septin7 protein was not examined in migrating myoblasts.

In non-migrating cells, the random filamentous structure of Septin7 was detected within the whole area of cells (Figure 1A). Control (non-transfected) cells, and cells transfected with a non-targeting scrambled sequence of shRNA (Scr) revealed highly organized Septin7 filaments, which were co-localized with actin (Appendix A). However, in KD cells (transfected with an shRNA plasmid targeting Septin7 sequence in the mRNA strand) a fragmented, pointwise appearance of the Septin7 was observed (Figure 1A). In addition, the actin network was also modified (Appendix A). On the other hand, this filamentous appearance was altered during migration, and well-organized structural units were formed in Ctrl and Scr cultures (Figure 1B) mainly around the nuclei and within the projections. The re-organization of septin filaments in migrating cells is more obvious in Appendix A, which shows that its original co-localization with actin was modified in relation to that of the non-migrating samples. In migrating KD cells, Septin7 filaments could be detected in cell projections, which showed co-localization with actin (Appendix A), whereas the previously described less-organized, pointwise Septin7 labeling was present in the remaining “stable” part of the cells.

The synthesis of Septin7-N-mCherry fluorescent protein from a eukaryotic expression vector and its intracellular pattern was followed by live cell imaging experiments (Figure 2). The newly formed Septin7 filaments showed a filamentous appearance (Figure 2A), which could be distinguished from the endogenous filaments using anti-Septin7-specific immunostaining (Figure 2B). Confocal images clearly indicate the specificity of the primary antibody recognizing Septin7 filaments expressed endogenously (green) and transcribed from the expression vector (red); the merged image represents co-localization of the two signals. The formation of Septin7-N-mCherry proteins followed in C2C12 myoblasts via imaging their presence in every 0.20 µm of the analyzed cell (Appendix A). Moreover, the successful expression of the fusion protein was also monitored in migrating C2C12 myoblasts (Figure 2C, Appendix A).

To summarize the findings above, a re-arrangement of the Septin7 filaments could be detected in migrating cells, which supports its structural role in the determination of cell morphology and front–rear polarity allowing the cells to move in different directions.

### 3.2. Migration of Myogenic Cells Is Accompanied by Changes in Intracellular [Ca^2+^]

Migration was initiated in myoblast cultures by a wound scratch (see Materials and methods), where non-migrating and migrating cells could clearly be distinguished (Figure 3A,B). The intracellular [Ca^2+^] of cells in migrating and non-migrating areas was investigated in Scr, and Septin7 KD cultures. In non-migrating cells, no significant difference between Scr and KD cells was detected in [Ca^2+^]_i_. However, a significant elevation of [Ca^2+^]_i_ was measured in migrating cells in both cultures (Figure 3C). In addition, [Ca^2+^]_i_ was significantly lower in migrating KD compared to that in the Scr cells. Elevated [Ca^2+^]_i_ levels were also measured in migrating Ctrl C2C12 myoblasts compared to those of their non-migrating counterparts (Appendix A); however, a significant difference between Ctrl and Scr cells was not detected either in migrating or non-migrating cells (Appendix A).

Based on these results, the elevation in intracellular calcium concentration has a significant role in cell migration, which is altered in C2C12 cultures with modified Septin7 expression.

### 3.3. Downregulation of Septin7 Protein Expression Altered the Migration of C2C12 Myoblasts

As the architecture of Septin7 filaments is distinct in migrating and non-migrating cells, the difference in the basic parameters of migration in the different C2C12 cell lines was determined. In these experiments, Mitomycin C, a potent DNA crosslinker was used to block cell division. For quantitative analysis, data were collected during the first 12 h from a 24 h long registration. A significant elevation of the length total way run (Figure 4A), average speed (Figure 4B), and maximum distance from origin (Figure 4C) were observed in Septin7 KD cells compared to those parameters of the Scr cultures. Next the two-dimensional displacement of the cells was determined every 20 min, and if this value did not exceed the size of the given cell, it was considered that the given cells did not move. The number of these occasions during the 12 h analyzed period was revealed to be significantly higher in Scr cultures compared to that of the KD cells (Figure 4D). Furthermore, representative trajectories of individual migrating cells indicated that KD cells took less turns during the first 4–5 h of migration (Figure 4F) compared to its Scr counterparts (Figure 4E). The total pathway was plotted as a function of time; thus, the significant difference between Scr and KD cells is illustrated in Appendix A. We could not detect a significant difference in the length total way run, average speed, and maximal distance from the origin between Ctrl and Scr samples (Appendix A).

Data above show that migration of Septin7 KD cells is severely modified as compared to that of Ctrl and Scr cultures, and KD cells are dedicated to moving in the direction they originally started.

### 3.4. FCF Stabilizes Septin7 Filaments in C2C12 Myoblasts

Recently, it was hypothesized that FCF might bind to the GDP binding site of the Septin7 protein [14]. In Figure 5B, the crystal structure of Septin7 in complex with GDP is presented, which might overlap with the FCF binding place for Septin7. The GDP binding site in the complete Septin7 dimer crystal structure is presented in Appendix A. The dose-dependent inhibition of the proliferation rate in C2C12 myoblasts by FCF was determined (Figure 5A), and according to the results an approximate IC_50_ concentration (100 µM) was used in the further experiments. Immunocytochemistry was performed following 24 h of FCF treatment in control C2C12 myoblasts, which revealed the altered architecture of Septin7 filaments. Longer and thicker filaments crossing the entire cytoplasm were observed in FCF-treated cultures (Figure 5C). In order to quantify the effect of FCF treatment, the thickness of individual Septin7 filaments was determined based on the width of green fluorescent filaments within the same size designated to the areas of the cells. Appendix A presents a significantly greater filament thickness in FCF treated cells compared to that of the non-treated controls. Since the confocal images represent a specific section/slice of the cell and septin filaments may extend into adjacent slices over or under the analyzed focus plane, filament length was not quantified. This experiment was repeated in cultures following transfection with the Septin7-N-mCherry plasmid, where the stabilizing effect of FCF on Septin7 filaments was also detectable in contrast to that of the non-treated cells (Figure 5D).

FCF thus effectively influences the assembly of Septin7 filaments; therefore, it should affect cell migration.

### 3.5. FCF Treatment Inhibits the Migration of C2C12 Myoblasts; S7-KD Cultures Are More Resistant to FCF

Since FCF treatment modifies the structure of Septin7 filaments and alters its dynamics in C2C12 cells, our next aim was to show its effect on the migration of myoblasts. FCF is capable to reversibly stabilize septin filaments within 3–4 h. The investigation of cell migration and the determination of basic parameters from these measurements were repeated in FCF-treated myoblast cultures. Changes in the length total way run (Figure 6A), average speed (Figure 6B), and maximal distance from the origin (Figure 6C) were calculated in Scr and S7-KD cultures as a percentage of non-treated values. All of these parameters were revealed to be significantly higher in KD cells compared to the Scr counterparts, indicating that FCF treatment was less effective in Septin7 KD cultures. To make them more visible, the length total way data were plotted as a function of time (Figure 6D,E), where the decrease initiated via FCF application was more pronounced in Scr samples. These results indicate that KD cultures with lower Septin7 protein expression are more resistant to FCF treatment.

This suggestion is also supported by representative trajectories of individual migrating cells from Scr and KD cultures, where the length of migration in KD cells during the first 4–5 h of migration (Appendix A) was altered less compared to that of the Scr counterparts (Appendix A). An alteration of cell migration in Ctrl cultures following FCF treatment is also clear (Appendix A), while the changes in parameters describing migration in FCF-treated Ctrl and Scr cells did not differ significantly (Figure 5D–F), further suggesting the similar behavior of these cultures regarding the cytokinin application and the significant deviation of KD cells from the control counterparts.

Our results confirm that alteration of either the expression (shRNA-based gene silencing) or the dynamical assembly (FCF treatment) of Septin7 filaments significantly modified the migration of myogenic cells. These results suggest the importance of cytoskeletal septin proteins in the process of myogenic development and muscle regeneration.

## 4. Discussion

### 4.1. Septin7 Filaments Are Important Cytoskeletal Parts of Myoblasts and Rearranged during Migration

From yeast to mammals, at all levels of phylogenesis, specific members of the septin family are an integral part of the cytoskeleton [43]. Among these, the Septin7 is unique and has a major contribution in the formation of septin oligomers and higher-order assemblies. In addition, *Sept7* deficiency causes embryonic lethality in mouse embryos, suggesting its vital role in embryogenesis. While a Septin7 KO mouse model does not exist, a muscle specific knockdown model has been established [41]. Similarly, C2C12 Septin7-KO cells are not viable due to the deficit in the cell cycle [41]. As Septin7 has a crucial role in different developmental statuses, its effect on cell migration could be an integral process to regulate the progression of different tissues. Here, we first reveal that the intracellular appearance and architecture of Septin7 are different in non-migrating and in migrating myoblasts. In non-migrating cells, Septin7 filaments are present all over the cytoplasm, while the more specific localization of filaments is detected during migration, especially in the cell projections. Even in KD cultures, migrating cells produce Septin7 filaments in the cell extensions; however, the number and size of these filaments are less extensive compared to those of the control and scrambled counterparts.

The role of Septin7 in the migration of different cell types has been reported, but the exact underlying mechanism, whether or not cellular specific environmental factors might alter the behavior of the protein, is still not clear [13,20,25,44,45]. It is well-known that the formation of distinct front and rear cellular areas are required for cellular polarization, which is important for directed migration and cell motility [46]. The regulatory effect of septin-related *cdc* genes (especially *cdc10*, the equivalent of mammalian *Sept7* in *S. cerevisiae* and in *Schizosaccharomyces pombe*) in determining cell polarity has also been examined [27,47]. Furthermore, the increased appearance of membrane blebbing, and the formation of long, thin appendages was detected in Septin7 KD T cells leading to diminished motility [17]. The emergence of an actin-mediated lamellipodial membrane protrusion forms the leading edge of the migrating cell, and the formation of the retracting tail is crucial. The special appearance of actin filaments was visible in all of our cell types; however, their intracellular distribution was also altered in KD cells, contributing to the round-shaped cell morphology. In Ctrl and Scr cells, Septin7 and actin co-localize and their borders are clearly distinguishable, while in KD cells the random distribution and pointwise appearance of Septin7 filaments seem to be less connected to the actin structure. All these aforementioned changes in Septin7 KD cells obviously define the different cell shapes and forms of cytoskeletal organization of migrating cells.

### 4.2. Modified Septin7 Expression Alters [Ca^2+^]_i_ Homeostasis in Migrating Myoblasts

The dynamic adaptation of intracellular calcium homeostasis in migration is essential in every cell type. An increasing front–rear [Ca^2+^]_i_ gradient is formed in migrating cells, having a role in the disassembly of focal adhesions, and the rear–end retraction and motility of the cell [46]. The front–rear polarity should be maintained during migration, because this biochemical process is required to block the formation of lamellipodia at the trailing edges [46,48,49]. Here, we show that resting [Ca^2+^]_i_ is not significantly different in Ctrl, Scr, and KD myoblasts; however, upon activation the migrating cells have increased [Ca^2+^]_i_, as it has been proven recently in C2C12 cells with a decreased expression of cell surface protein Syndecan-4 [50]. Interestingly, this elevation in [Ca^2+^]_i_ is significantly lower in migrating KD cells compared to that of their control counterparts, which might be related to Septin7 depletion. Several studies are dealing with the connection between septins and cytoplasmic [Ca^2+^]_i_ regulation. It has been shown that Septin7 regulates Ca^2+^ entry through Orai channels in human neural progenitor cells and neurons [37]. In addition, there is a verified connection between the overexpression of dseptin7 in neurons of wild-type *Drosophila* and modified calcium homeostasis leading to significant flight defects [51]. Taking the data together, our results suggest that Septin7 depletion altered intracellular calcium ion homeostasis, which might have regulated the movement and parameters of migration in cultured myoblasts. The exact mechanism underlying these alterations and mapping the related signaling pathways require further investigation.

### 4.3. Amount of Septin7 Regulate Migration Parameters in Myoblasts

Migration is a crucial process in which orchestrated signaling pathways direct the cells to the target place [1]. Rho-family small GTPases regulate migration, which has been validated in several studies so far [10,11,46]. Among the Rho GTPases, Cdc42 and its interacting partners are often analyzed together with septins. It has been reported on hematopoietic stem cells (HSCs), that the Cdc42–Borg4–Septin7 axis regulates the polarity of cells [52]. According to recent findings, the expression level of Septin7 might result in increased or decreased migration parameters. The overexpression of Septin7 caused the inhibition of migration in glioma cells [24], while the opposite effect prevailed in breast cancer cell lines [20]. Here, we present that Septin7 deficiency caused myoblasts to migrate faster and over longer distances compared to Scr cells. To resolve the contradiction of the effect of changes in Septin7 expression on cell mobility one needs to take into account the limitations of the different assay types used for analyzing cell migration [53]. Those include the size of the scratch and the injuries of the bordering cells in wound healing assays. Additionally, assays should not be longer than 24 h, because it is hard to decide whether closure of the gap is generated by cell proliferation and/or migration. In our experiments, we used a culturing insert to keep the bordering cells mostly untouched and cells were treated with Mitomycin C to block cell proliferation. In addition, in our experiments myoblasts could migrate in all directions. Different cell types can either migrate alone (myoblasts), in loosely connected populations (mesenchymal cells) or collectively as sheets of cells (such as epithelial cells) [53]. Although KD cells can migrate faster and over longer distances, additionally spending more time migrating, these cells are not able to form viable and intact myotubes, as reported previously [41]. Several regulatory proteins (transcription and myogenic regulatory factors) that control the fusion of myoblasts into myotubes are probably involved in this alteration. In a recent study, the importance of alternative splicing and the dynamic variability of the Mef2D transcription factor in the process of myogenic differentiation were described [54]. In addition, several signaling pathways that are involved in skeletal myogenesis, such as the p38α/β mitogen-activated protein kinase (MAPK) pathway in which the Rho family of small GTPases including Rac1, Cdc42, and RhoA [55] could also play a role. The analysis of through which molecules exactly and at which point of the differentiation the whole process is modified in Septin7 KD cells awaits further investigations. Based on the above, we can conclude that Septin7 has a role in the migration of myoblasts indicating its importance in muscle myogenesis and regeneration. It should also be mentioned that in vivo myoblasts need to migrate and penetrate through the basement membrane and connective tissue barriers, the perimysium or endomysium [9]. As these biological circumstances were not simulated in our experiments, present data provide the basis for further experiments regarding the exact regulatory function of Septin7 in myoblast migration in vivo.

### 4.4. Dynamic Rearrangement of Septin Assembly Is Required for Myoblast Migration

The molecular mechanism underlying heteromeric and higher-order septin complexes is still a controversial area of septin biology [7,13,56,57]. After revealing the first crystal structure of septin hexamer (SEPT7-SEPT6-SEPT2-SEPT2-SEPT6-SEPT7), this is often referred as a canonical complex within the cells [7,57]. In this arrangement, the G interface is free at each end. It has also been suggested that any given septin proteins from the same structural group substitutes another in the complex [7]. According to predictions, 20 viable hexamers and 60 octamers could exist; however, the generality of this theory has not been fully established [58]. As a consequence, more and more data suggest that the septin literature and structural assumptions need to be critically evaluated [58]. Analyzing septin complexes, Septin2 and Septin7 have been suggested as core molecules. Since the reconstruction of higher-order septin assemblies is quite dynamic, all possibilities that could affect this process are the focus of septin research. FCF is a plant cytokinin and has the ability to dampen septin dynamics [14,59]. In silico docking experiments revealed that nucleotide-binding pockets of septins are the assumed binding place for FCF (see Figure 5B). Our data are comparable to those published earlier, as FCF treatment negatively affected cell proliferation and decreased the speed and length of total way run of migrating C2C12 myoblasts (Figure 6 and Appendix A). However, off-target effects have also been published with FCF and septins [29,59]; its efficacy on Septin7 stabilization seems to be specific, since mCherry-tagged Septin7 reacted the same way for FCF treatment (as shown in Figure 5D). The effects of FCF are mainly attributed to its ability to affect septin distribution, assembly, dynamics, or interactions with other proteins, such as ErbB2/HER2 [60], without affecting either actin or tubulin polymerization [30].

## 5. Conclusions

In conclusion, we provide evidence that Septin7 filaments are rearranged during the migration of myoblasts. The downregulation of Septin7 expression resulted in altered parameters of migration; KD myoblasts travelled faster and further compared to those of the non-modified cultures. The explanation of these findings could not only be the lack of valuable septin filaments, but also the significantly smaller changes in the intracellular calcium concentration of migrating KD myoblasts. In addition, Septin7 knockdown cells seem to use more dedicated routes during migration instead of changing their direction of movement as often as we saw in scrambled cultures. Furthermore, the special role of Septin7 in the migration of myoblasts was also confirmed via FCF treatment, where the dynamic rearrangement of septin filaments was inhibited, causing a more pronounced alteration of migration in Scr cells compared to KD cultures. A graphical summary of our results is presented in Figure 7. Altogether, Septin7 protein has an essential role in the highly orchestrated process of migration; thus, it is crucial in the regeneration and development of functional skeletal muscle fibers.

## Figures and Tables

**Figure 1 cells-12-01825-f001:**
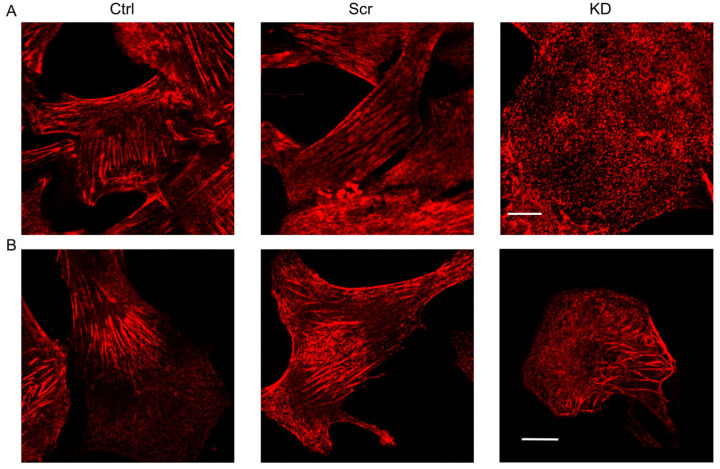
Immunofluorescent detection of Septin7 in non-migrating and migrating myoblasts. Cytoskeletal filaments containing Septin7 were probed with an anti-Septin7 antibody and the appropriate Cy3-labeled secondary antibody (red) in control (Ctrl), scrambled shRNA-transfected (Scr) and Septin7-knockdown (KD) C2C12 myoblasts. Immunofluorescent signals from cells within the non-migrating (**A**) and migrating (**B**) region of the individual cultures were detected at a high resolution using the Zeiss Airyscan confocal microscope with a 40× oil immersion objective and 535 nm excitation wavelength. Scale bars represent 10 µm.

**Figure 2 cells-12-01825-f002:**
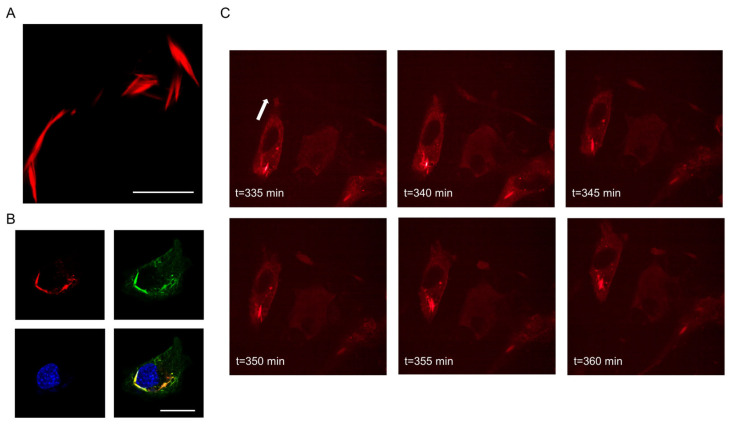
Expression of a fluorophore-tagged Septin7 protein and its filamentous appearance in C2C12 myoblasts during migration. Exogenous mCherry-tagged Septin7 expression was induced from a eukaryotic expression vector in proliferating C2C12 myoblasts. The intracellular distribution of the fluorophore-labeled Septin7 filaments (red) was followed by measurements using a confocal laser scanning microscope. (**A**) Fluorescent image of an mCherry-N-Septin7 fusion protein in C2C12 myoblasts. Scale bar is 10 µm. (**B**) Localization and distribution of mCherry-N-Septin7 fusion protein (red) and endogenously synthesized Septin7 (green) following immunolabeling. The Septin7 antibody (green) was used to label all proteins within the cells. The merged image represents the intracellular localization of the exogenous and endogenous Septin7 proteins. Nuclei were labeled with DAPI (blue). Scale bar is 20 µm. (**C**) Series of images from migrating C2C12 myoblasts. Exogenous mCherry-N-Septin7 filaments were followed by Opera Phenix measuring setup during the migration of myoblasts, and images were taken every 5 min showing the position of cells and the actual structure of Septin7 filaments. The white arrow indicates the direction of movement of a given cell.

**Figure 3 cells-12-01825-f003:**
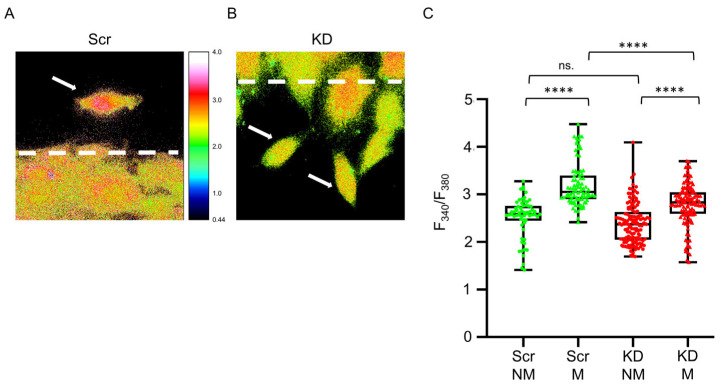
Changes in intracellular [Ca^2+^] in migrating C2C12 cultures. Intracellular calcium concentration was determined in Fura-2 AM-loaded Scr (green) and Septin7-KD (red) myoblasts. (**A**) Representative ratiometric images from Fura-2-loaded Scr (**A**) and KD (**B**) cultures. Dashed lines represent the manually defined boundary line between non-migrating and migrating regions of the cell cultures. White arrows label migrating cells. Mind that the color scale indicates different fluorescence ratios measured at 340 and 380 nm; a warmer color represents a higher intracellular calcium concentration. (**C**) Fluorescence ratio measured in non-migrating (NM) and migrating (M) Scr (green) and KD cells (red). Statistics were calculated via a one-way ANOVA and Bonferroni’s multiple comparison test (**** *p* < 0.001; *n* = 62 and 86 cells in non-migrating and migrating Scr samples; and *n* = 130 and 125 cells in non-migrating and migrating KD samples, respectively, from Three independent experiments). Here, and in all subsequent figures the rectangles in the box plots present the median and the 25 and 75 percentile values, while the error bars point to 1 and 99%.

**Figure 4 cells-12-01825-f004:**
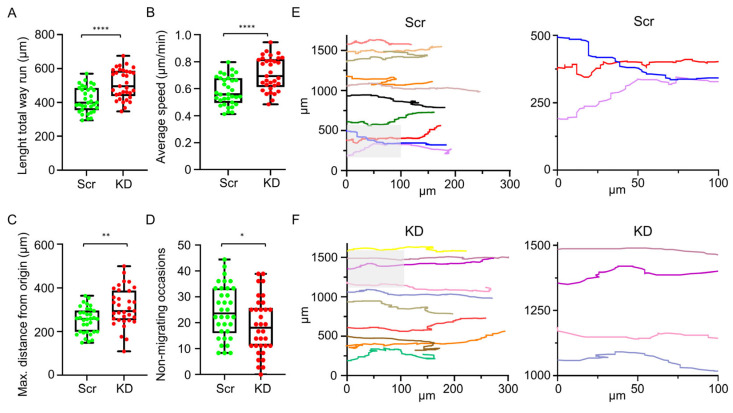
Alteration in the parameters of migration in Septin7 knockdown cultures. Parameters describing the properties of cell migration were determined in the Scr (green) and KD (red) cultures. 12 h of migration was analyzed and the length of total way run (**A**), average speed (**B**), and maximum distance from the original position of the individual cells (**C**) were calculated. The number of occasions within 12 h of measurement where the movement of cells in any direction did not reach the size of cells (evaluated as non-migrating section) was determined from a 20 min long period of original recordings (**D**). Statistics were calculated via a one-way Anova and Bonferroni’s multiple comparison test (* *p* < 0.05, ** <0.01, **** <0.001; *n* = 34 cells in Scr, and *n* = 33 cells in KD, respectively, from 3 independent experiments). Representative trajectories of migrating Scr (**E**) and KD (**F**) C2C12 myoblasts are presented (left panels), where different colors identify individual cells during the first 12 h of the migration. To enable a more detailed visualization of the trajectories, panels G and H present enlarged regions of panels E and F, respectively, as indicated by the gray boxes.

**Figure 5 cells-12-01825-f005:**
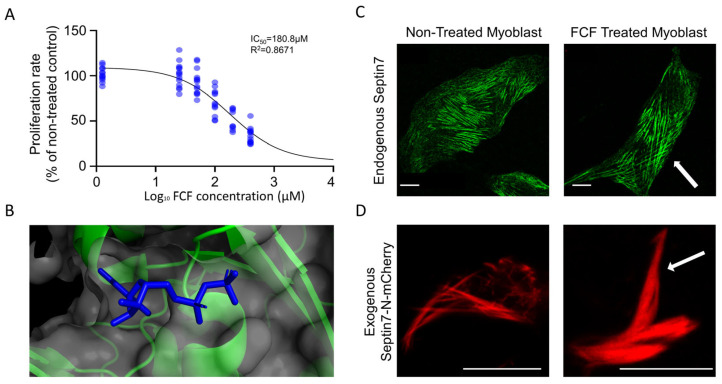
Forchlorfenuron (FCF) treatment stabilizes the Septin7 filaments and modifies cell proliferation. (**A**) Effect of different concentrations of FCF on the proliferation of C2C12 myoblasts determined after 24 h of treatment. Data points are presented as normalized to the non-treated cultures and were fitted with the Hill equation resulting in an IC_50_ value of around 181 μM. (**B**) Assumed binding place for FCF in Septin7 protein, based on docking experiments, see reference in text. Septin7 (PDB code: 3T5D) is used to visualize the GDP-bound state of the Septin7 complex. FCF is supposed to interact with the binding place of GDP (blue). Forchlorfenuron treatment stabilizing septin polymers generating thicker and longer Septin7 filaments (indicated by white arrows) both in control C2C12 cells (**C**), and in myoblasts expressing exogenous mCherry-tagged Septin7 (**D**). Scale bars represent 10 µm.

**Figure 6 cells-12-01825-f006:**
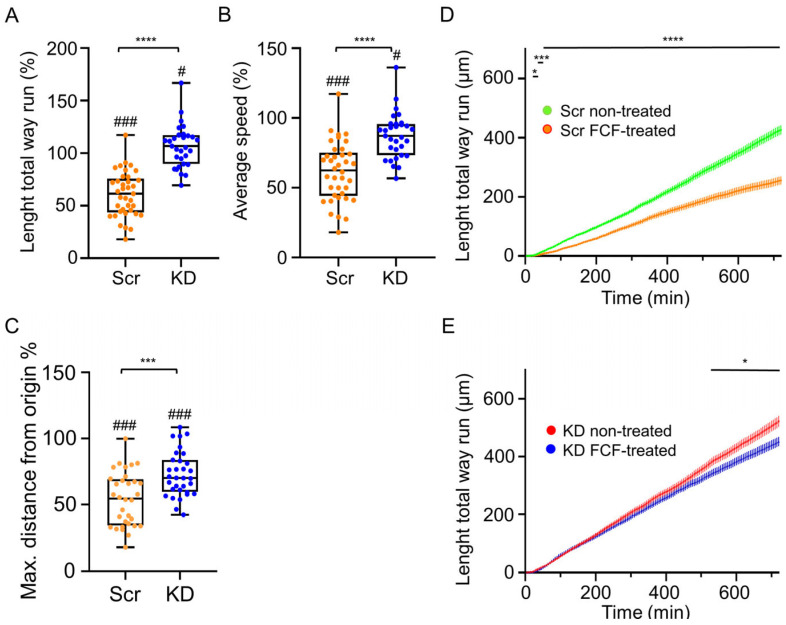
FCF treatment alters the parameters of migration with C2C12 Septin7 KD cells being more resistant. Length total way run (**A**), average speed (**B**), and maximum distance from origin (**C**) of FCF-treated Scr (orange) and KD (blue) cultures determined during 12 h of migration, and parameters of FCF-treated cells presented as a percentage values of non-treated cells. Statistics were calculated via an ordinary one-way ANOVA and Bonferroni’s multiple comparison test (* *p* < 0.05, *** < 0.005, **** < 0.001 (comparing the FCF-treated Scr and KD cell data), # < 0.05, and ### < 0.001 (comparing the non-treated and FCF-treated cultures); *n* = 37 cells in Scr, and *n* = 32 cells in KD, respectively, from 3 independent experiments). To better illustrate the changes following FCF treatment, individual points of length total way run in Scr cells (**D**) and KD myoblasts (**E**) were plotted as a function of time with and without FCF treatment.

**Figure 7 cells-12-01825-f007:**
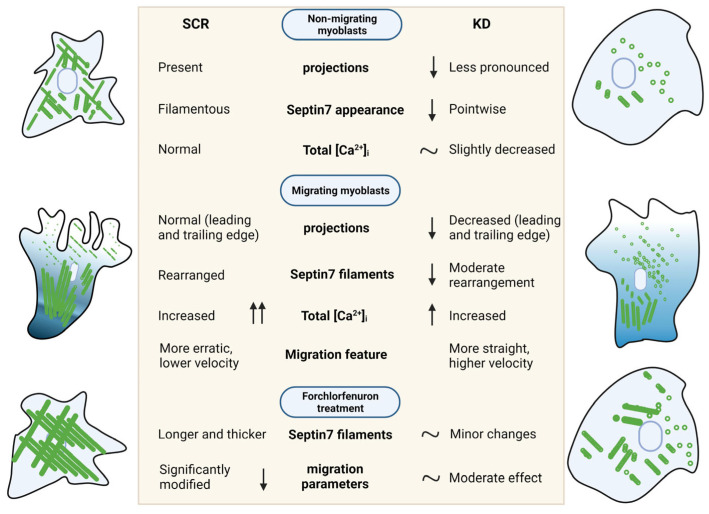
Graphical summary. Contribution of Septin7 protein in morphological, intracellular and cytoskeletal alterations in the migration of myoblast. Figure was created with BioRender.com, accessed on 18 June 2023.

## Data Availability

The data presented in this study are available on request from the corresponding author. The data are not publicly available due to the fact that most of the data, pictures and videos require specific softwares or equipments to open or handle.

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
