# Peer review of "Migration of Myogenic Cells Is Highly Influenced by Cytoskeletal Septin7"

_cells, 2023, doi:10.3390/cells12141825_

Round 1

Reviewer 1 Report

General concerns:

The manuscript is interesting, however, its novelty is limited.  It confirms both the role of septin 7 in cell polarity regulation as well as the role of calcium in cell motility. Both are well known. Authors have already shown the crucial role of septin7 in the C2C12 cell line differentiation. The study's main result is the effect of septin7 on cell motility; it is a piece of important knowledge. Especially, since there are reports, showing the positive role of septins on the T lymphocyte ameboid motility. This should be discussed taking into account cell motility characteristics in each cell type.

Specific concerns:

Human septins are coded by 13 separate genes, so we can not describe them as “isoforms”.

Septin7 is highly similar to the yeast CDC10 protein, so it is suspected to be a cell polarity regulator for a long time.

It is long and well-known, that the cytoplasmic calcium in motile cells is highly labile and connected to the cytoskeleton contraction cycle as well as cell adhesion (Lee J et al. 1999, Pomorski P et al. 2004). Do studied cells also perform calcium transients during motility? What influence has a septin7 on the rate of calcium transients? The single measurement is not a proper measurement here.

No comments

Reviewer 2 Report

The authors described the research “Migration of myogenic cells is highly influenced by cytoskeletal Septin7.” that are essential for  the migration of myoblasts implying its contribution to muscle myogenesis and regeneration.  

This is an interesting study in an area that needs investigating. This is also a carefully written manuscript, and the findings are of considerable interest. 

A several minor revisions are listed below.

Comments:

In the Introduction (line 97-), the authors should cite any papers describing the effects of Forchlorfenuron on the cytoskeletal organization. The authors should also provide additional details on the reason for using this drug in this experiment.

In Figure 1, the scale bar should be in FIgure  B KD only. Because, all scale bars are the same size. The blank space between photos is useless. The space between pictures is useless, so you should make the pictures larger.

Figure 2 B is a montage image of each staining? Please separate into individually. This picture is not clear about the border. C says cells in migration, please indicate the migrating part with an arrow or arrowhead.

Please separate into individually. Are they different cells? If they are the same cells, they should be combined into one picture. I am not sure what you are trying to say, especially in picture D. A little more magnification might help. Photos C and D have different magnification. What are they comparing?

Ca2+, 37oC, and CO2 should be accurately shown using upper case characters or lower case characters in all over the manuscript.

Please provide a schematic illustration of what is being said in the Discussion or Conclusion.

Ca2+, 37oC, and CO2 should be accurately shown using upper case characters or lower case characters in all over the manuscript.

Line278 "Intracellular distribution of the fluor-278 ophore-labeled Septin7 filaments (red) was followed by a confocal microscope." The authors use  confocal laser scanning microscopy? This sentence is wrong.
